# Quantifying Impacts of Mean Annual Lake Bottom Temperature on Talik Development and Permafrost Degradation below Expanding Thermokarst Lakes on the Qinghai–Tibet Plateau

**Feng Ling [1,2,\*] and Feifei Pan [2]**

[1] School of Mathematics and Statistics, Zhaoqing University, Zhaoqing 526061, China
[2] Department of Geography and the Environment, University of North Texas, Denton, TX 76203, USA; feifei.pan@unt.edu
[\*] Correspondence: lingf@zqu.edu.cn

**Abstract:** Variations in thermokarst lake area, lake water depth, lake age, air temperature, permafrost condition, and other environmental variables could have important influences on the mean annual lake bottom temperature (MALBT) and thus affect the ground thermal regime and talik development beneath the lakes through their direct impacts on the MALBT. A lake expanding model was employed for examining the impacts of variations in the MALBT on talik development and permafrost degradation beneath expanding thermokarst lakes in the Beiluhe Basin on the Qinghai–Tibetan Plateau (QTP). All required boundary and initial conditions and model parameters were determined based on field measurements. Four simulation cases were conducted with different respective fitting sinusoidal functions of the MALBTs at 3.75 °C, 4.5 °C, 5.25 °C, and 6.0 °C. The simulated results show that for lakes with MALBTs of 3.75 °C, 4.5 °C, 5.25 °C, and 6.0 °C, the maximum thicknesses of bowl-shaped talik below the lakes at year 300 were 27.2 m, 29.6 m, 32.0 m, and 34.4 m; funnel-shaped open taliks formed beneath the lakes at years 451, 411, 382, and 356 after the formation of thermokarst lakes, with mean downward thaw rates of 9.1 m/year, 10.2 m/year, 11.2 m/year, and 12.0 m/year, respectively. Increases in the MALBT from 3.75 °C to 4.52 °C, 4.25 °C to 5.25 °C, and 5.25 °C to 6.0 °C respectively resulted in the permafrost with a horizontal distance to lake centerline less than or equal to 45 m thawing completely 36 years, 32 years, and 24 years in advance, and the maximum ground temperature increases at a depth of 40 m below the lakes at year 600 ranged from 2.16 °C to 2.80 °C, 3.57 °C, and 4.09 °C, depending on the MALBT. The ground temperature increases of more than 0.5 °C at a depth of 40 m in year 600 occurred as far as 74.9 m, 87.2 m, 97.8 m, and 106.6 m from the lake centerlines. The simulation results also show that changes in the MALBT almost have no impact on the open talik lateral progress rate, although the minimum distances from the open talik profile to lake centerlines below the lakes with different MALBTs exhibited substantial differences.

**Keywords:** mean annual lake bottom temperature (MALBT); open talik; permafrost degradation; numerical simulation; Qinghai–Tibetan Plateau (QTP)

## 1. Introduction

Thermokarst lakes are a unique natural part of the landscape of the high-latitude and high-altitude regions where ice-rich permafrost exists. The impacts of thermokarst lakes on the permafrost thermal regime and underlying talik progress have been widely studied by many researchers during the past several decades because thermokarst lakes are (1) important heat sources for the ground surrounding the lakes [1–7]; (2) major producers of methane if taliks develop under the lakes [8–12]; and (3)

a sensitive indicator of climatic and environmental changes [13–19]. To study thermokarst lakes, the numerical simulation approach is probably one of the most effective tools because numerical modeling can provide otherwise unavailable insights into the long-term evolution of the ground thermal regime and talik development beneath thermokarst lakes [2,20–28].

The development processes of taliks beneath thermokarst lakes in the Alaskan arctic and the Qinghai–Tibetan Plateau (QTP) are quite different. Open taliks usually cannot form below thermokarst lakes in the Alaskan Arctic region [2,3,29], while they can form under thermokarst lakes in the QTP after long-term influence. Furthermore, thermokarst lakes usually do not drain or shrink after the formation of open taliks in the QTP due to the low water permeability of the lake bottom sediment and the underlying weathered mudstone, and permafrost lateral thaw would continue, even if the underlying permafrost had thawed completely [25,26,30].

The long-term impacts of the QTP's thermokarst lakes on the spatial and temporal variability of the permafrost thermal regimen and talik development have been studied numerically in recent years. The lake bottom temperature is usually assumed to vary according to a sinusoidal function with a mean annual lake bottom temperature (MALBT) in the numerical model. On the basis of the environmental and climate conditions of a monitored thermokarst lake in the Beiluhe Basin on the QTP, Ling et al. investigated the rate of talik development beneath the lake and the rate of lateral thaw of permafrost after an open talik had formed. The MALBTs of the sinusoidal function were assumed to be 5.2 °C in the central deep pool and 4.22 °C in the shallow nearshore pool [31]. Li et al. developed and applied a coupled heat–moisture numerical model to simulate the temperature and moisture processes of the permafrost around a non-penetrative thermokarst lake in the Beiluhe Basin for analyzing the influence of the thermokarst lakes on the permafrost under global warming. The MALBT in the sinusoidal function was set to be 5.5 °C [25]. Based on the moving mesh method, Wen et al. developed a dynamic model of lake growth with phase change and investigated the morphologic processes of a thermokarst lake and its long-term influence on the permafrost thermal regimen in the Beiluhe Basin using a MALBT of 2.0 °C [26]. All these studies indicate that the MALBT is a critical variable for modeling the permafrost thermal regime and talik development beneath thermokarst lakes.

The MALBT is a product of air temperature, snow thickness on the top of the lake ice and its properties, lake ice thickness, solar radiation, thermal characteristics of the water body and the adjacent ground, size and depth of the lake, and the composition and history of accumulation of bottom sediments. Variations in the variables mentioned above could result in changes in the MALBT, and then, in turn, they can affect talik development and the ground thermal regimen underneath the lakes. Therefore, it is necessary to quantify the impact of variations in the MALBT on the talik development beneath thermokarst lakes in permafrost regions. This is especially important for the QTP: First, the QTP plays important roles in the regional environmental system, the Asian monsoon system, the global atmospheric circulation and water cycle, and global climate change. Second, in addition to the existence of more than 1500 thermokarst lakes in the QTP with different areas and depths [32], new thermokarst lakes have been forming since a mass of small pits were generated on the ground surface when some infrastructure projects such as the Qinghai–Tibetan Railway, fiber-optic cables, and oil pipelines were constructed in the past 20 years. These small pits have subsequently been filled with water and eventually developed into thermokarst lakes, which in turn caused serious thermal erosion and thaw settlement and continuously produced impacts on permafrost ecosystems and the environment in the QTP [30,33]. Finally, there is abundant evidence showing accelerating permafrost degradation in the QTP in recent decades owing to global warming [34–37].

On the basis of observations and previous studies in the Beiluhe Basin, the objective of this study is to apply the numerical modeling approach to assess the impact of variations in the MALBT on (1) talik development and open talik formation time, (2) the open talik lateral progress process, and (3) permafrost degradation beneath thermokarst lakes.

## 2. Study Area Details

The Beiluhe Basin is located in the hinterland of the QTP. The elevation of the region is above 4500 m. Permafrost in this basin has been observed and studied since the 1950s. Several representative thermokarst lakes have been monitored since 2006 by the State Key Laboratory of Frozen Soil Engineering, Chinese Academy of Sciences; their locations are shown in Figure 1. Based on the previous studies [35–39], the characteristics of permafrost and thermokarst lakes in the basin can be summarized as follows: (1) This area is underlain by continuous ice-rich permafrost. The permafrost thickness is about 20 m to 80 m, the mean annual permafrost temperature ranges from −2.0 °C to −0.58 °C, and the geothermal gradient varies from 0.015 °C m$^{-1}$ to 0.048 °C m$^{-1}$. (2) The active layer is located in 1.8 m to 3.0 m, and the ice-rich permafrost is located at depths from 1.8 m to 6.0 m with a thickness of about 3.0 m. (3) The permafrost thaw settlement coefficient is about 60%, and the permafrost temperature at a depth of 15 m varies from −1.6 °C to −0.9 °C. (4) More than 30 thermokarst lakes have formed in the lower parts of the Beiluhe Basin where ice-rich permafrost or massive ground ice exists. (5) The majority of the lakes are elliptical in shape with a maximum major axis of more than 150 m and a minimum minor axis of less than 20 m. (6) The lake water depth ranges from 0.5 m to 2.5 m, the lake ice thickness in the winter varies from 0.45 m to 0.7 m, and the lakeshore collapsed during late September and early October, with a slumping width of 0.0 to 1.8 m.

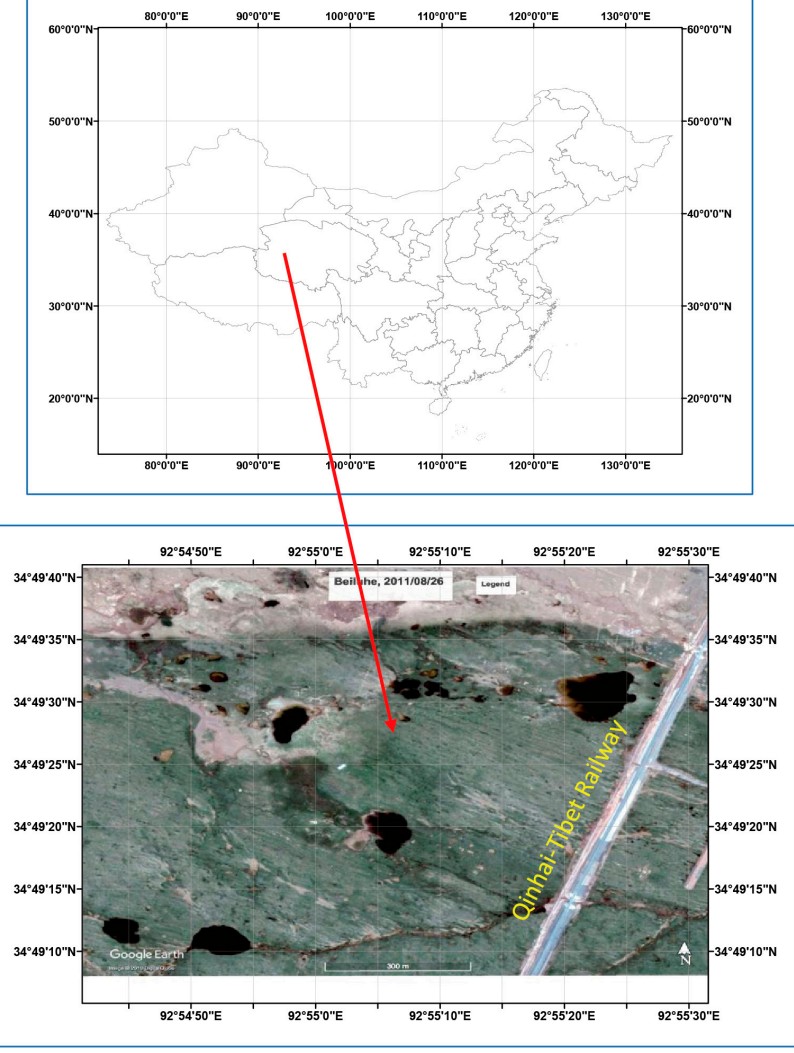

**Figure 1.** The locations of the monitored representative thermokarst lakes in the Beiluhe Basin on the Qinghai–Tibetan Plateau.

## 3. Model Description

### 3.1. Lake Radius Expanding Model

A simple lake radius expanding model was employed to simulate the thermal regimen of permafrost and talik development under thermokarst lakes in the Beiluhe Basin on the QTP in this study. This finite element model was modified from a well-tuned two-dimensional heat transfer model with phase change in a cylindrical coordinate system [2] by forcing the intermittent thermokarst lake radius expansion as an external influence [40]. Neglecting water migration in the system, the governing equations describing the thermal regimen of talik and permafrost underneath an expanding thermokarst lake can be written in the following form:

$$C\frac{\partial T}{\partial t} = \frac{\partial}{\partial r}\left(k\frac{\partial T}{\partial r}\right) + \frac{k}{r}\frac{\partial T}{\partial r} + \frac{\partial}{\partial z}\left(k\frac{\partial T}{\partial z}\right),$$
$$(0 < t < D, 0 < z < Z, 0 < r < R).$$

(1)

$$C = \begin{cases} C_f, & T < T_e - \Delta T, \\ C_f + L_w\rho_b\frac{W-W_u}{\Delta T}, & T_e - \Delta T \leq T \leq T_e \\ C_u, & T > T_e. \end{cases}$$

(2)

$$k = \begin{cases} k_f, & T < T_e - \Delta T, \\ k_f + \frac{k_u-k_f}{\Delta T}[T - (T_e - \Delta T)], & T_e - \Delta T \leq T \leq T_e \\ k_u, & T > T_e. \end{cases}$$

(3)

$$r(t) = r_0 + v_{he}t$$

(4)

where $T$ is temperature in °C; $t$ is time in seconds; $z$ is the depth from the ground surface downward in m; $Z$ and $R$ are the total depth and radius of the analysis domain in m; $D$ is the total simulation period in seconds; $C$ and $k$ are the volumetric heat capacity in J m$^{-3}$ °C$^{-1}$ and the thermal conductivity in W m$^{-1}$ °C$^{-1}$; the subscripts $f$ and $u$ are the frozen and unfrozen phases, respectively; $T_e$ is the thawing or freezing temperature and is set at 0 °C; $\Delta T$ is the temperature interval in which the phase change occurs in °C and is assumed to be 1.0 °C; $[T_e - \Delta T, T_e]$ is the temperature range in which the phase change occurs; $Lw$ is the mass specific latent heat of freezing in J kg$^{-1}$; $\rho_b$ is the dry buck density of soil in kg m$^{-3}$; $W$ is the total water content percent of soil by mass; $r$ and $r_0$ are the radius from the lake centerline at time $t$ and the initial radius of the lake in m; and $v_{he}$ is the mean lake radius lateral expansion rate in m/year.

### 3.2. Thermokarst Lake Expansion

Natural thermokarst lakes expand their basin margins by thawing ground ice and redistributing the sediment. The lake radius increases gradually by lake shoreline retreat via complex mechanical and thermal erosion processes [41,42]. Several studies have shown that the long-term mean lake lateral expansion rate can be simply described as a linear relation between the lake radius and time [18,26,43]. Based on the field investigations and research results in the Beiluhe Basin [26,30,43], the initial radius of the lake formed over the ground surface was set to be $r_0 = 6.0$ m, and the mean lake radius lateral expansion rate which increased linearly in Equation (4) was set to be $v_{he} = 0.25$ m/year. Because lake radius expansion caused by lake bank slumping and retreat is a transient and intermittent process, rather than a gradual and continuous process, the mean lake radius lateral expansion rate of 0.25 m/year was treated as a stepwise change, i.e., the lakeshore collapsed and the ground surface boundary transformed into the lake bottom boundary instantaneously on 30 September every four years, with a lake radius increment of 1.0 m after each collapse.

This study focuses on the sensitivity of talik development and permafrost degradation under thermokarst lakes to variations in the MALBT. The lake depth was set to be a constant determined by the depth of the ice-rich permafrost thickness and permafrost thaw settlement coefficient. The

ice-poor permafrost below the ice-rich permafrost was assumed to have no thaw settlement occurrence. No attempt was made to accommodate basin deepening.

### 3.3. Analysis Domain and Physical and Thermal Parameters

The centerline of the expanding lake was selected as the symmetric axis (Figure 2). The radius of the analysis domain was 220 m from the lake centerline, which is more than 64 m away from the maximum distance from the lakeshore of the expanding thermokarst lake in the simulation period to reduce the influence of lateral boundary conditions on the simulation results. The permafrost thickness was 50 m, and the mean annual temperature at the depth of 15 m was −1.2 °C. The depth of the computational domain was 90 m from the ground surface, which is 40 m below the permafrost base to reduce the effect of bottom boundary conditions. The ice-rich permafrost was assumed to locate in depths from 2.3 m to 5.3 m, with a permafrost thaw settlement coefficient of 0.6. Based on previous studies [31,40], the soil column in the analysis domain was divided into four layers, and their physical and thermal parameters are listed in Table 1. Because of the axial symmetric nature, only half of the domain was simulated in this study.

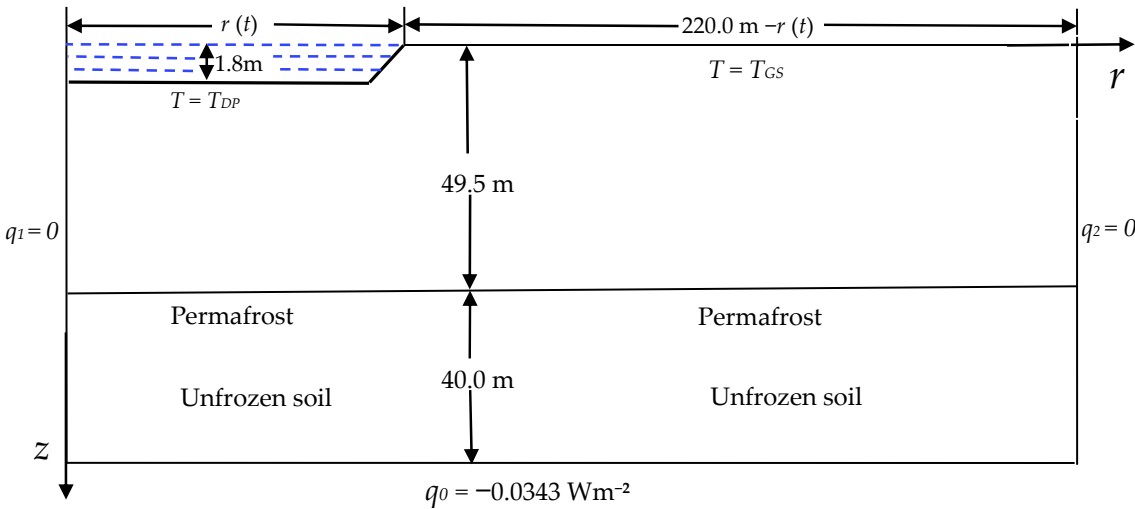

**Figure 2.** Diagrammatic drawing of the analysis domain. The lake radius expanded from the initial length of 6.0 m with a horizontal expansion rate of 0.25 m/year. The upper boundaries are set at a depth of 0.5 m below the lake bottom off the shore and at a depth of 0.5 m beneath the ground surface on the shore. The soil types in the analysis domain are silty clay, silt and clay, sand and gravel, and mudstone successively from the upper boundary to the lower boundary; the corresponding soil thicknesses are 2.3 m, 3.0 m, 9.7 m, and 75.0 m, respectively.

**Table 1.** The soil types and their physical and thermal parameters used in this study.

| No. | Range (m) | Soil Type | Dry Bulk Density (kg m$^{-3}$) | Water Content by Mass (kg kg$^{-1}$) | Unfrozen Water Content by Mass (kg kg$^{-1}$) | Frozen Thermal Conductivity (W m$^{-1}$ °C) | Thaw Thermal Conductivity (W m$^{-1}$ °C) | Frozen Volumetric Heat Capacity (J m$^{-3}$ °C) | Thaw Volumetric Heat Capacity (J m$^{-3}$ °C) |
|---|---|---|---|---|---|---|---|---|---|
| I | $0.0 \leq z \leq 2.3$ $R_0 \leq r \leq 220.0$ | Silty clay | 1400 | 0.35 | 0.04 | 2.28 | 1.18 | $2.371 \times 10^6$ | $3.220 \times 10^6$ |
| II | $2.3 < z \leq 5.3$ $0 \leq r < 220$ | Silt and clay | 1400 | 0.56 | 0.06 | 2.12 | 1.42 | $2.543 \times 10^6$ | $3.450 \times 10^6$ |
| III | $5.3 < z \leq 15.0$ $0 \leq r \leq 220.0$ | Sand and gravel | 1600 | 0.27 | 0.03 | 1.92 | 1.35 | $2.459 \times 10^6$ | $3.178 \times 10^6$ |
| IV | $15.0 < z \leq 90.0$ $0 \leq r \leq 220.0$ | Mudstone | 1600 | 0.17 | 0.02 | 1.57 | 1.28 | $1.872 \times 10^6$ | $2.457 \times 10^6$ |

### 3.4. Boundary Conditions and Model Solution

The upper boundary of the analysis domain was set at a depth of 0.5 m below the lake bottom off the shore and at a depth of 0.5 m below the ground surface on the shore to avoid the difficulties of determining the heat and physical parameters in the complicated composition of subsurface soils [31]. The upper boundary temperature condition was approximated by a sinusoidal function as follows:

$$T = T_a + A \, \sin\left(\frac{2\pi}{8760}t - \varphi\right) \tag{5}$$

where $T_a$ is the mean annual ground temperature on the upper boundary, $A$ and $\varphi$ denote the corresponding amplitude of temperature fluctuation and initial phase angle, and $t$ presents time in hours. Based on the ground temperature measurements and previous studies in the Beiluhe Basin [30,31,34,41], $T_a$ (which is actually the MALBT) was set to be 3.75 °C, 4.5 °C, 5.25 °C, and 6.0 °C; $A$ and $\varphi$ were assumed to be 3.15 °C and $-3\pi/4$, respectively. The sinusoidal functions with four different MALBTs for representing the lake bottom boundary conditions are shown in Figure 3. The lake shore upper boundary condition is represented by the following sinusoidal function:

$$T_{GT} = -1.22 + 7.82\sin\left(\frac{2\pi}{8760}t - \frac{3\pi}{4}\right)(0 \leq t \leq D, z = 0.5, R(t) \leq r \leq 220). \tag{6}$$

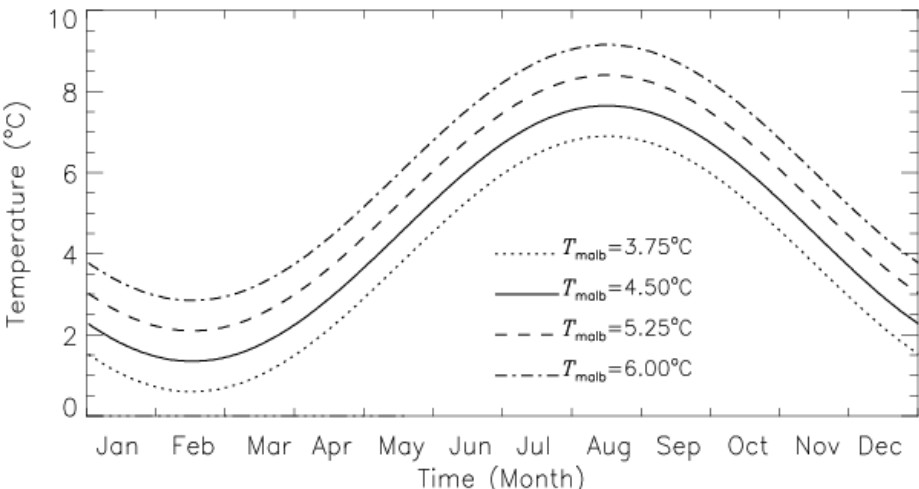

**Figure 3.** The sinusoidal functions with four mean annual lake bottom temperatures ($T_{\text{malb}}$). These are the upper boundary conditions of the lake bottom.

A constant heat flux was set to be the lower boundary condition as follows:

$$-\frac{\partial T(Z,r,t)}{\partial z} = q \ (0 \leq t \leq D, z = 90, 0 \leq r \leq 220). \tag{7}$$

The vertical boundary along $r = 0$ had no horizontal heat flux due to geometric symmetry, the vertical boundary along $r = 220$ m was regarded as an infinite boundary, and the lateral boundary conditions were treated as no horizontal heat flux along both vertical boundaries:

$$\frac{\partial T(z,r,t)}{\partial r} = 0 (0 < t < D, 2.3 < z < Z, r = 0) \tag{8}$$

$$\frac{\partial T(z,r,t)}{\partial r} = 0 (0 < t < D, 0.5 < z < Z, r = 220) \tag{9}$$

where $q$ is the geothermal heat flux in W m$^{-2}$, and $q = -0.0538$ W m$^{-2}$ was calculated from the observed geothermal gradient of 0.0343 °C m$^{-1}$ on the permafrost base and the corresponding soil heat conductivity listed in Table 1.

The governing Equations (1)–(9) were solved from one four-year thaw slumping period to another with a time step of one day using a combination of the finite difference method and Galerkin's finite element method [40,44] until the open taliks had been formed for more than 150 years below the bottoms of the lakes with MALBTs of 3.75 °C, 4.5 °C, 5.25 °C, and 6.0 °C. The permafrost thickness was assumed to be 50 m, and there was no thermokarst lake over the permafrost initially. The model was run using the parameters presented in Table 1 with $R(t) = 0$ and the boundary conditions given by Equations (6)–(9) on 17 April when the mean permafrost surface temperature was −1.202 °C, until the equilibrium state of the soil thermal regimen was formed with the corresponding physical and thermal parameters and boundary conditions. The calculated ground temperature field was then used as the initial condition, and a small thermokarst pond with a radius of $R(t) = 6$ m and a depth of $H_0 = 1.8$ m was assumed to be over the permafrost (Figure 2). The radius of the small expanding lake increased by 1.0 m on 30 September every 4 years during the whole simulation period. Three-node triangular elements were used with the spatial step varying from 0.9 m to 1.5 m along the $z$-axis direction and from 1.0 m to 3 m along the $r$-axis direction. The numerical model was coded in Fortran PowerStation 4.0.

## 4. Results and Analyses

### 4.1. Talik Development

The simulated ground thermal regimens and talik thicknesses beneath the expanding thermokarst lakes with MALBTs of 3.75 °C, 4.5 °C, 5.25 °C, and 6.0 °C on the last days of years 300 and 450 after the thermokarst lakes' formation are presented in Figure 4. Due to the strong heat source effect, a bowl-shaped talik formed under each thermokarst lake, and talik thickness increased gradually with time. The maximum talik thicknesses below the lakes under these four different MALBTs were 27.2 m, 29.6 m, 32.0 m, and 34.4 m, respectively, on the last day of year 300 (Figure 4a–d). Accompanied by a continuous increase in the talik thickness, an open talik formed underneath each thermokarst lake after a relative long period, and the open talik volume increased with time. On the last day of year 450 after the formation of the thermokarst lakes, a funnel-shaped open talik started to form beneath the lake when the MALBT was 3.75 °C (Figure 4e). Funnel-shaped open taliks formed beneath the lake when the MALBTs were 4.5 °C, 5.25 °C, and 6.0 °C, and the corresponding minimum horizontal distances from the open talik to the lake centerline were 36.7 m, 50.0 m, and 59.2 m (Figure 4f–h). An increase in MALBT can lead to increases in the permafrost downward thaw rate and decreases in the open talik formation time beneath the lake. The open talik formation times below the thermokarst lakes ranged from 451 years to 356 years, and the corresponding permafrost mean downward thaw rates varied from 9.1 cm/year to 12.0 cm/year (Table 2). Increasing the mean lake bottom temperature by 0.75 °C in the Beiluhe Basin led to shortening the open talik formation time by 26 years to 40 years and increasing the permafrost mean downward thaw rate by 0.8 cm/year to 1.2 cm/year.

**Table 2.** The open talik formation times and corresponding permafrost mean downward thaw rates below thermokarst lakes with different mean annual lake bottom temperatures.

| Mean annual lake bottom temperature (°C) | 3.75 | 4.5 | 5.25 | 6.0 |
|---|---|---|---|---|
| Open talik formation time (year) | 451 | 411 | 382 | 356 |
| Permafrost mean downward thaw rate (cm/year$^{-1}$) | 9.1 | 10.2 | 11.2 | 12.0 |

Figure 5 shows the vertical profiles of the open taliks underneath thermokarst lakes with different MALBTs on the last day of year 540 after the thermokarst lakes' formation. Increases in the MALBT from 3.75 °C to 4.5 °C, 5.25 °C, and 6.0 °C caused an increase in the minimum distance from open talik profiles to the lake centerlines from 58.7 m to 72.9 m, 83.6 m, and 90.1 m, respectively. To estimate the impact extent of variations in the MALBT on open talik development, Table 3 presents the minimum distances from open talik profiles to lake centerlines below the lakes with different MALBTs at different years after the thermokarst lakes' formation, and the results listed in Table 3 indicate that the difference in the minimum distance between MALBTs of 3.75 °C and 6.0 °C decreased from 39.2 m in year 480 to 28.5 m in year 600.

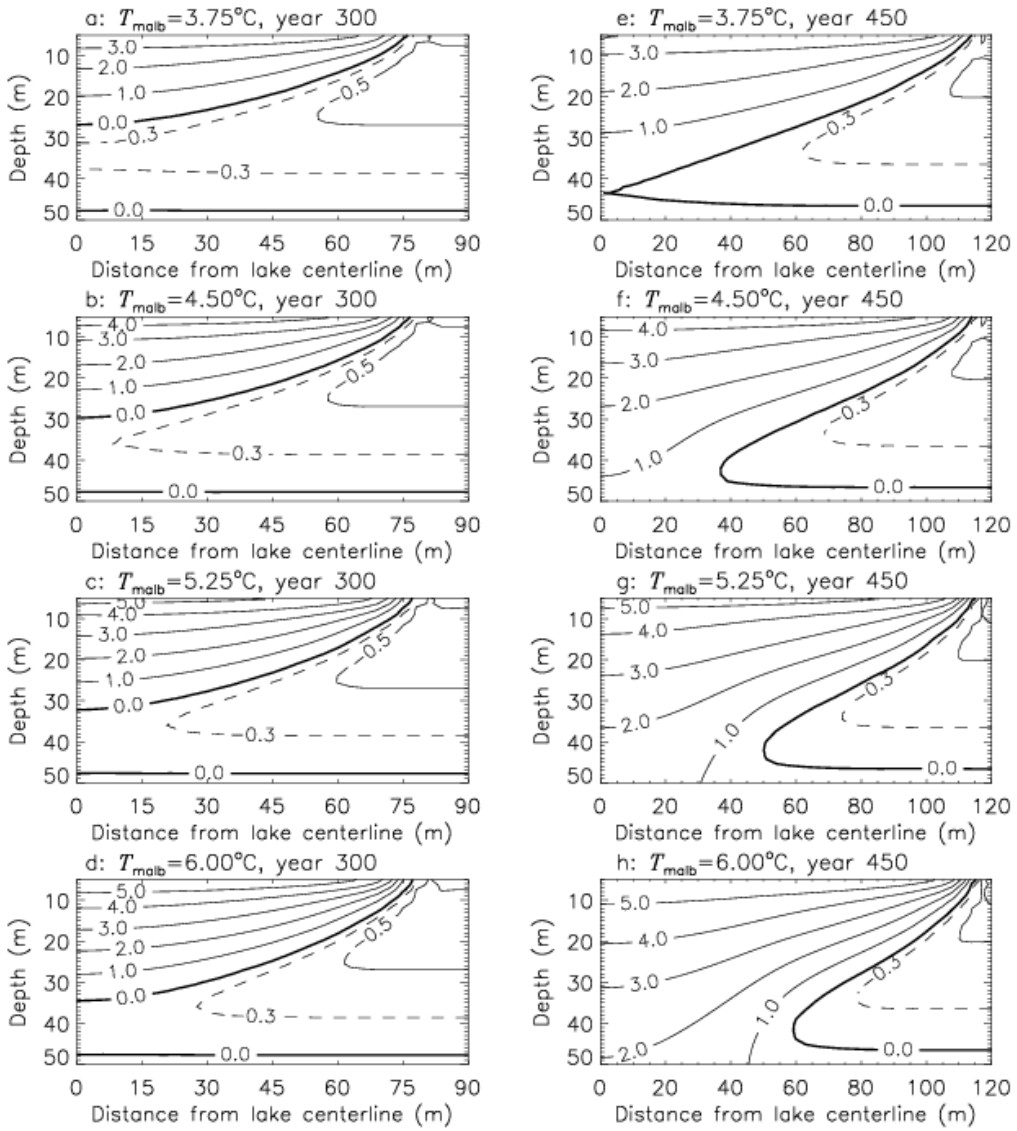

**Figure 4.** Simulated ground thermal regimens and talik thicknesses beneath the expanding thermokarst lakes with different mean annual lake bottom temperatures on the last days of year 300 (**a–d**) and year 450 (**e–h**) after the thermokarst lakes' formation.

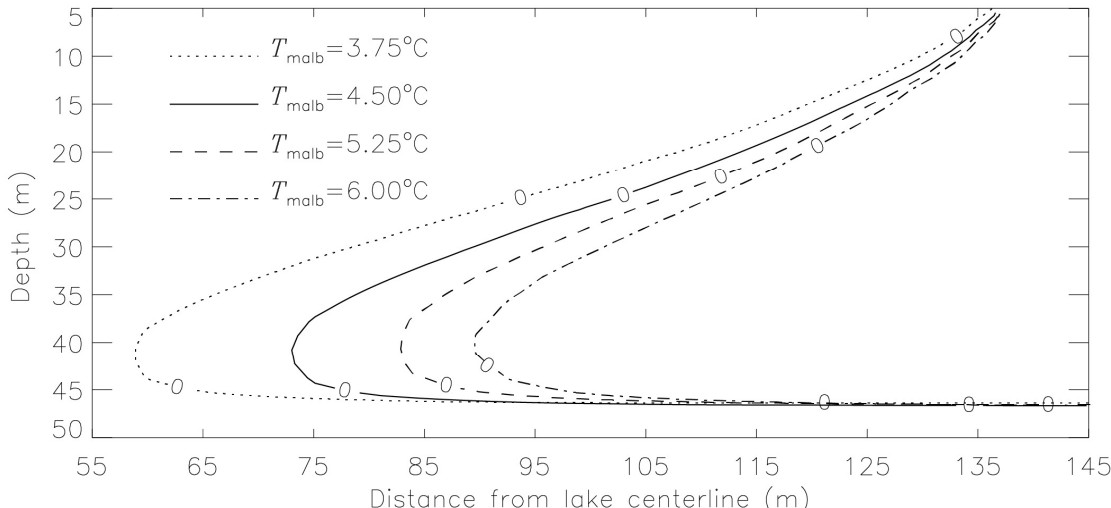

**Figure 5.** The vertical profiles of the open talik below the thermokarst lakes with different mean annual lake bottom temperatures on the last day of year 540 after the thermokarst lakes' formation.

**Table 3.** The minimum distances from open talik profiles to lake centerlines below the lakes with different mean annual lake bottom temperatures at different years.

| Mean Lake Bottom Temperature (°C) | The Minimum Distance at Different Years (m) | | |
|:---:|:---:|:---:|:---:|
| | **480** | **540** | **600** |
| 3.75 | 31.9 | 58.7 | 81.3 |
| 4.50 | 50.8 | 72.9 | 92.5 |
| 5.25 | 62.2 | 83.6 | 102.4 |
| 6.00 | 71.1 | 90.1 | 109.8 |

In order to fully understand the difference in open talik development processes beneath thermokarst lakes with different MALBTs, the mean open talik lateral progress rate, $R_{mlp}$, in m/year was defined as a measure index in this study:

$$R_{mlp} = \frac{d_{\min}}{t_{otc}} \tag{10}$$

where $d_{min}$ is the minimum distance from the open talik profile to the lake centerline in m, and $t_{otc}$ is the time interval between the timings of the downward and upward advancing 0 °C isotherms starting to merge related to the current time in years.

To explore the impact of variations in the MALBT on open talik development with time, Figure 6 presents the vertical profiles of the open taliks below thermokarst lakes with MALBTs of 3.75 °C, 4.5 °C, 5.25 °C, and 6.0 °C on the last day of years 45, 90, and 150 after the open taliks' formation.

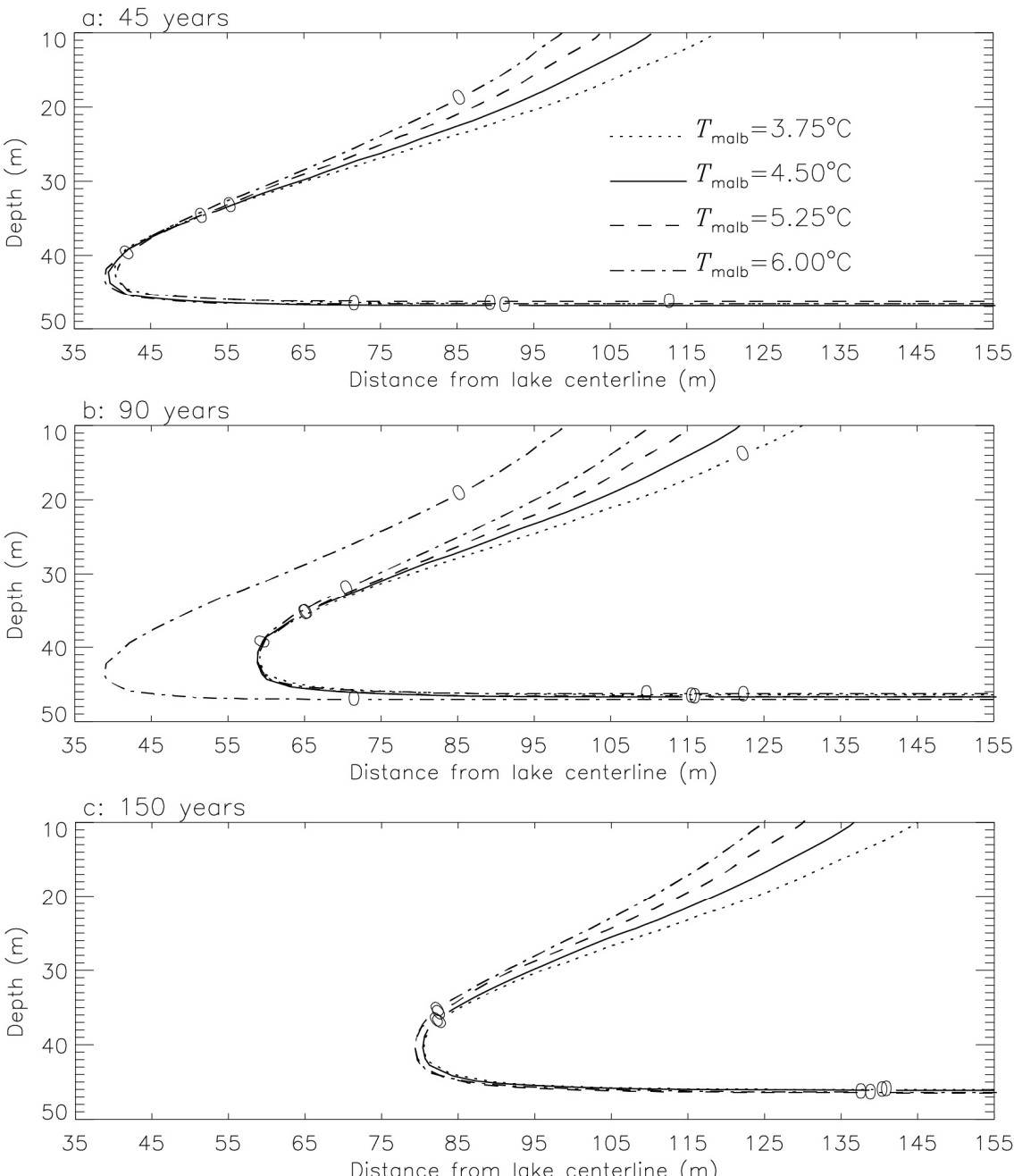

**Figure 6.** The vertical profiles of the taliks below the thermokarst lakes with different mean annual lake bottom temperatures on the last day of years (**a**) 45, (**b**) 90, and (**c**) 150 after each open talik formation.

Significantly, open taliks beneath thermokarst lakes with different MALBTs have the same mean open talik lateral progress rates of 8.4 m/year, 6.4 m/year, and 5.3 m/year at years 45, 90, and 150 after the open taliks' formation. These results suggest that variations in the MALBT have an insignificant influence on open talik lateral progress. This is due to the fact that thermokarst lakes with lower lake bottom temperatures had longer lake radii when the open talik formed, and the ground under the lake thus had a higher temperature because of the heat source effect of lake water. For example, 150 years after open talik formation, the minimum distances from the open talik profile to the lake centerline below the four lakes were all about 80.0 m; the thermokarst lakes with MALBTs of 3.75 °C, 4.5 °C, 5.25 °C, and 6.0 °C formed for 613 years, 573 years, 544 years, and 518 years, and the corresponding lake radii reached 159 m, 149 m, 142 m, and 135 m, respectively.

### 4.2. Permafrost Degradation

Figure 7 shows permafrost degradation processes during the period from when the radius of lakes with different MALBTs expanded from 43 m to the time when the permafrost along $r$ = 45 m thawed completely. Permafrost along $r$ = 45 m thawed from the permafrost table downward and from the permafrost base upward with thermokarst lake expansions. For MALBTs of 3.75 °C, 4.5 °C, 5.25 °C, and 6.0 °C, permafrost along $r$ = 45 m thawed completely at years 497 (Figure 7a), 461 (Figure 7b), 429 (Figure 7c), and 405 (Figure 7d), respectively. Increasing the MALBT by 0.75 °C resulted in permafrost along $r$ = 45 m thawing completely in advance for 24 to 36 years, depending on the MALBT. Increases in the MALBT could speed up the permafrost degradation process below the lake. Figure 7 also indicates that the permafrost downward thaw rate was much higher than the upward thaw rate, and the four downward and upward advancing 0 °C isotherms below the lakes all merged at a depth of about 40 m.

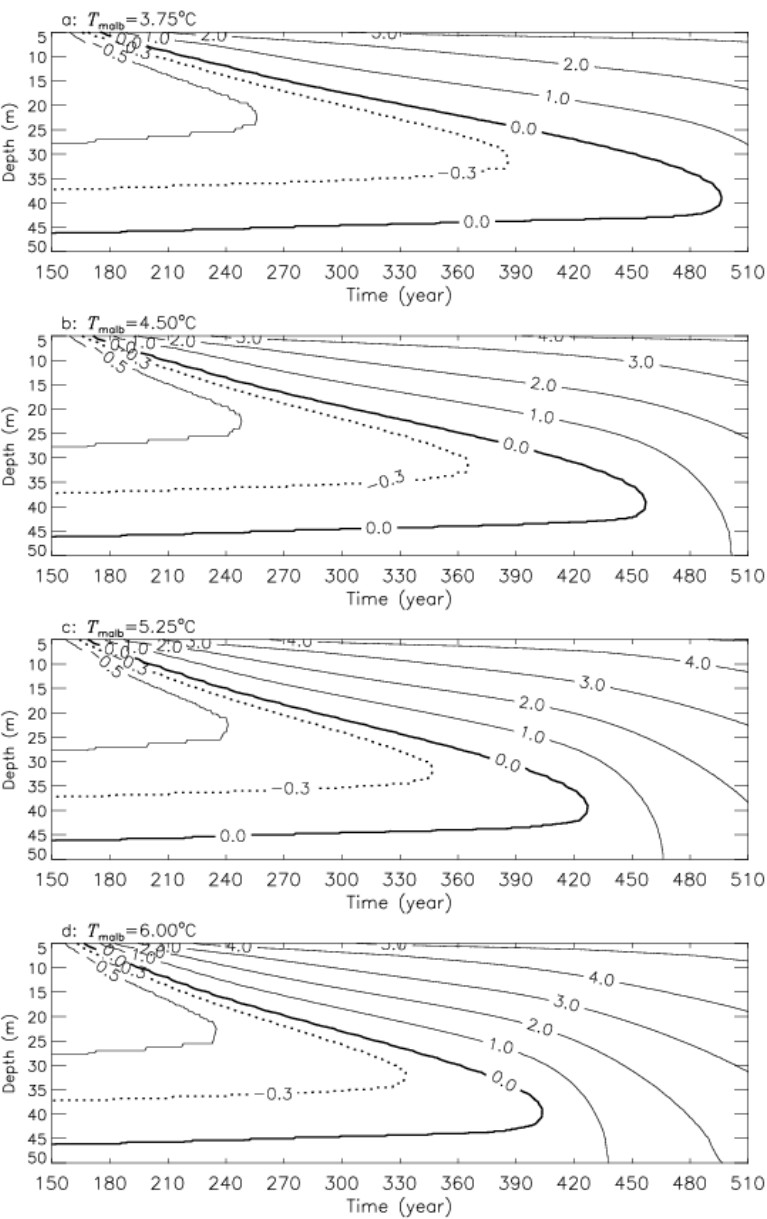

**Figure 7.** Permafrost degradation processes along $r$ = 45 m nearby the expanding thermokarst lakes with the mean annual lake bottom temperatures (MALBT) of (**a**) 3.75 °C, (**b**) 4.50 °C, (**c**) 5.25 °C, and (**d**) 6.00 °C.

To further compare the differences in permafrost degradation processes underneath thermokarst lakes with different MALBTs, Figure 8 presents the ground temperature horizontal profiles along depth $z$ = 40 m beneath the thermokarst lakes with different MALBTs at year 600 after the lakes' formation. Increases in the MALBT from 3.75 °C to 6.0 °C with an increment of 0.75 °C for 600 years caused increases in the maximum ground temperature from 2.16 °C to 2.80 °C, 3.57 °C, and 4.09 °C and increases in the ground temperature by more than 0.5 °C at distances of 74.9 m, 87.2 m, 97.8m, and 106.6 m respectively, from the lake centerlines.

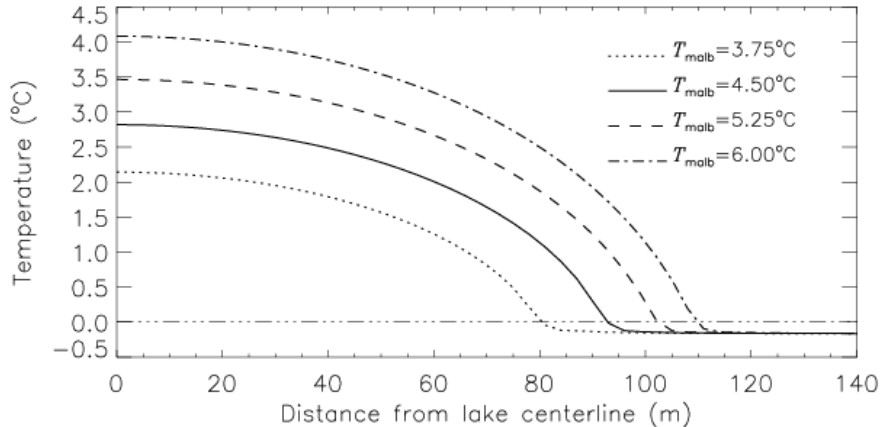

**Figure 8.** The horizontal profiles of ground temperature along depth $x$ = 40 m beneath the thermokarst lakes with different long-term mean lake bottom temperatures at year 600 after each thermokarst lake formation.

## 5. Discussion

The simulated results in this study suggest that talik development beneath a thermokarst lake is sensitive to the MALBT, and increases in the MALBT can speed up talik development below the thermokarst lakes and accelerate permafrost degradation. Previous studies showed that talik development has a significant influence on the physical, mechanical, chemical, ecological, geomorphological, and biological processes occurring in the surrounding thermokarst lakes: (1) Taliks provide an anaerobic environment wherein bacteria decompose organic matter formerly preserved in permafrost. Accelerated talik development will boost methane flux into the atmosphere, contribute a further influence to local climate change, and exacerbate global warming [8–11,45]. (2) Permafrost degradation caused by talik development leads to increases in lake depth and area due to sequential thaw settlement and bank retreat, resulting in changes in both surface and subsurface water storage. This alters the hydrological balance in the ground near the thermokarst lakes and seriously affects the regional ecosystems [25,46–49]. (3) Talik development leads to temperature rise in the adjacent permafrost and increase of the active layer thickness, reduces the mechanical stability of the permafrost to support a load, and seriously affects the performance of structures constructed in permafrost regions [16,49,50]. Therefore, increasing the MALBT would inevitably deteriorate the local environmental system, provide positive feedback to local and global climate change [51,52], and even give rise to some hazards.

Permafrost in the QTP is characterized by relatively thin thickness, high ground temperature, and high ice content [33–35]. These features make the permafrost extremely sensitive to environmental change, climate change, and anthropogenic activities. More than 1500 lakes exist in the QTP at present [31,32]. This abundant surface feature is the result of local environmental disturbances and permafrost degradation. To prevent natural hazards caused by increases in the MALBT in the QTP, some measures should be adopted. On the one hand, in order to avoid the potential influence of thermokarst lakes on the heat stability of infrastructure in the future, a high level of attention should be paid to the heat erosion effect of thermokarst lakes during the site selection and design period. On the other hand, surface disturbances that can induce increases in the MALBT should be minimized during

the infrastructure construction and maintenance stages by preventing increasing lake depth, removing small pits on the ground surface, and trying to avoid destroying surface vegetation.

This study quantifies the impact of variations in the MALBT on talik development beneath the lake and permafrost lateral thaw progress after open talik formation below thermokarst lakes. Since accurate relations among lake depth, the MALBT, and air temperature and the MALBT are hard to obtain [53,54], sinusoidal functions with MALBTs of 3.75 °C, 4.5 °C, 5.25 °C, and 6.0 °C were used to represent the lake bottom upper boundary conditions, while the corresponding amplitude of temperature fluctuation and the thermokarst lake lateral expansion rate were assumed to be constant. We should emphasize that we should be cautious about the above two assumptions. First, thermokarst lake expansion is affected by many factors, and variations in the MALBT and other factors such as permafrost condition and formation history, lake water depth, and lake size and age may all lead to changes in the lake radius expansion rate. Second, variations in the MALBT may also result in changes in the amplitude of temperature fluctuation in Equation (5). Finally, main water sources feeding lakes, precipitation falling into lakes, surface and subsurface inflows, and permafrost meltwater affect the lake water balance, but so do the heat transfer and exchange in the lake [55], which are not considered in the numerical model used in this study. In reality, there are some problems for which one or even two of the above assumptions may not be physically meaningful. However, from our point of view, such assumptions enable us to focus on the key impact of the MALBT. In order to accurately quantify the impact of thermokarst lakes on the permafrost thermal regimen and talik progress, an accurate MALBT and mean annual lake lateral expansion rate estimated from systematic, comprehensive, and long-term field measurement data are clearly necessary.

## 6. Summary and Conclusions

Based on previous studies of thermokarst lakes and permafrost in the Beiluhe Basin in the QTP, the influence of variations in the MALBT on talik development and permafrost lateral thaw progress after the formation of the open talik was studied through a series of numerical simulations using a lake expanding model. The following conclusions can be drawn from the simulation results:

(1) An increase of 0.75 °C in the MALBT can advance the formation time of a funnel-shaped open talik by more than 26 years in the Beiluhe Basin.

(2) The minimum distance from the open talik boundary to the thermokarst lake centerline is very sensitive to the MALBT, while the MALBT plays a less important role in the open talik lateral thaw process.

(3) Increases in the MALBT can speed up the permafrost degradation rate below the lake.

(4) Selections of accurate MALBT and mean annual lake lateral expansion rate based on long-term field measurement data are prerequisites for numerical simulation.

**Author Contributions:** F.L. designed the idea and outline of the article, developed the software used in this work, conducted the simulations, and wrote the manuscript. F.P. revised the original manuscript. F.L. and F.P. plotted the figures and approved the revised manuscript.

**Funding:** This study was supported by the National Natural Science Foundation of China (Grant no. 41271076) and the Natural Science Foundation of Guangdong province, China (Grant no. 2015A030313704).

**Acknowledgments:** We would like to express our gratitude to the two anonymous reviewers for their constructive suggestions and insightful comments to improve this manuscript.

**Conflicts of Interest:** The authors declare no conflict of interest. Financial support does not constitute an endorsement of the views expressed in this paper.

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
