# Peer review of "Quantifying Impacts of Mean Annual Lake Bottom Temperature on Talik Development and Permafrost Degradation below Expanding Thermokarst Lakes on the Qinghai–Tibet Plateau"

_water, doi:10.3390/w11040706_

Round 1

Reviewer 1 Report

This manuscript presents new and interesting data and findings.

However, the presentation of the material has some flaws and I have the following suggestions for possible improvements:

- The theme could be presented in a wider and global context. Accordingly, the Introduction section and especially the Discussion section could be expanded by highlighting the importance of the theme and the findings more clearly and in a global context.

- The Discussion section should also be clearly expanded by direct comparisons with pulications on the theme from other study areas worldwide.

- In the Discussion and Conclusions sections some more information on wider implications of the interesting findings should be provided and discussed.

- Please add a scale and a N-arrow in Figure 1. 

Author Response

Response to comments from Reviewer #1

 I uploaded my reply as a Word file.

Reviewer 2 Report

The paper was nice to read, as it also tackles an important problem, i.e. the permafrost melt and lake / talik development in current permafrost region. The authors have presented their work well, and it was clear to read. I am not a mathematician, so I was not probably best for reading the equations. In any case, the equations and methods seemed sound. However, my comments relate to the other parts of the paper, and not the equations. Here are the detailed comments:

1. Abstract, lines 26-28: I do not understand the sentence ”Increases in the MALBT from 3.75 to 4.526 , 5.25, and 6.0 resulted in the permafrost with a horizontal distance to lake centerline less 27 than or equal to 45 m thawed completely in 24 years to 36 years in advance,” Is the increment, which is mentioned here, between 3.74->4.526 or 4.526 ->5.25 or 5.25->6 or 3.75->6? And what are the increments in distances between each of these temperature intervals? Please, add values, and clarify, what do you actually mean? This has been explained better in the results sections, thus please clarify it also here.

2. Figure 2: Please, mark in the figure, which part is the lake/pond? Is it the one where the area on the top left corner, where there is 1.8 m depth arrows marked? please calrify the figure. Also it would be good to add there, where the soil zones are. Now it is impossible to read, how deep each of the zones presented in Table 1 go in the Figure 2.

3. Table 1: Does the range values start from the bottom of the lake, or from the land surface next to the lake? please clarify. And also add these soil type ranges in the Figure 2. Which soil types are within the permafrost, and which are below the permafrost base (49.5 m) presented in Figure 2?

4. Figure 4: You mention the years 300, 40 etc. in the Figure 4. However, these temporal simulation procedures do not come clear from the main text. Therefore, please clarify in the methods sections, which temporal simulations scales were applied in which simulations and for which purposes? It seems based on the figures, that there were many different temporal scales, but it is not clear based on the text. Clarificaiton is needed.

5. Figure 6: In this figure the years 45-150 years are presented. Please, calrify, what temporal scales and why were selected for the analyses. Add this in the methods section more clearly.

6. Page 10 line 293: you mention “…open talik just formation…” -> What does the word “just2 mean there? Is it mistake? if not, pelase clarify.

Author Response

I uploaded my reply as a word file.

Round 2

Reviewer 1 Report

The authors have carefully addressed all comments, suggestions and requests of the reviewers and have improved their manuscript accordingly.